# Patient Experience from a Pilot Study Implementing Software-Based Post-COVID Case Management in GP Practices—A Qualitative Process Evaluation

**DOI:** 10.3390/healthcare13141701

**Published:** 2025-07-15

**Authors:** Kathrin Sesterheim, Frank Peters-Klimm, Annika Baldauf, Charlotte Ullrich, Uta Merle, Joachim Szecsenyi, Sandra Stengel

**Affiliations:** 1Department of Primary Care and Health Services Research, Heidelberg University Hospital, Heidelberg University, Im Neuenheimer Feld 130, 69120 Heidelberg, Germany; kathrin.sesterheim@med.uni-heidelberg.de (K.S.); frank.peters-klimm@med.uni-heidelberg.de (F.P.-K.); annika.baldauf@med.uni-heidelberg.de (A.B.); charlotte.ullrich@med.uni-heidelberg.de (C.U.); joachim.szecsenyi@med.uni-heidelberg.de (J.S.); 2Department of Internal Medicine IV, Heidelberg University Hospital, Heidelberg University, Im Neuenheimer Feld 410, 69120 Heidelberg, Germany; uta.merle@med.uni-heidelberg.de

**Keywords:** post-COVID, general practitioners, patients, case management

## Abstract

**Background/Objectives**: In Germany, the provision of healthcare for post-COVID patients primarily lies with general practitioners (GPs), who often lack the necessary knowledge and skills. As part of the PostCovidCare pilot study (PCC), case management software incorporating a symptom diary was introduced and piloted in *n* = 10 GP practices with *n* = 33 included patients involved (September 2022–March 2023). This study aimed to explore patients’ experiences. **Methods**: Semi-structured telephone interviews were transcribed and analyzed using qualitative content analysis. A total of *n* = 10 patient interviews were conducted (July–September 2023). **Results**: Patients’ experiences were heterogeneous. The service was largely structured, involving an extensive initial assessment, follow-up appointments, questionnaires, and support from medical assistants, but technical problems with the symptom diary occurred. The GP consultation played a prominent role. Positive aspects included being actively asked about their symptoms, being given a lot of time, initiating diagnostic and therapeutic measures, and having a closer relationship with their GP. Negative aspects included the time taken, resulting exhaustion, duplication of efforts, and insufficient involvement in the consultation process. **Conclusions**: The pilot study conducted at an early stage of the post-COVID era demonstrated the basic feasibility of case management in primary care from patients’ perspectives. In addition, for future projects, it is important to integrate patients into the design from the outset, adapt the software to users’ needs, and consider care providers’ perspectives.

## 1. Introduction

The SARS-CoV-2 virus triggered a global pandemic in March 2020, which did not become endemic until 2023 [1]. By May 2024, the World Health Organization (WHO) had counted 775,552,205 reported SARS-CoV-2 infections [2]. After SARS-CoV-2 infection, the majority of individuals recover [3]. However, some patients develop persistent symptoms, from which some may recover and some may suffer severe functional impairment [3]. 

The term “long-COVID” was first mentioned by Callard and Perego [4] and is now used for cases where symptoms may persist for more than four weeks [5]. Post-COVID syndrome is defined by the WHO [6] as follows: "Post COVID-19 condition occurs in individuals with a history of probable or confirmed SARS-CoV-2 infection, usually 3 months from the onset of COVID-19 with symptoms that last for at least 2 months and cannot be explained by an alternative diagnosis".

In total, there are more than 50 symptoms that can be associated with post-COVID that affect daily life [7,8]. The most common symptoms include fatigue, shortness of breath (dyspnea), cognitive dysfunction (e.g., brain fog), depression, joint pain (arthralgia), cough, chest pain, and persistent loss of smell and taste [6,7,9,10,11,12]. 

The prevalence of post-COVID is highly heterogeneous [13], in part due to differences in methodological approaches, follow-up periods, and different definitions of post-COVID [7]. In their population-based study from a German federal state, Peter et al. [14] reported a prevalence of post-COVID of 6.5% in the adult population.

### 1.1. Current State of Health Care for Post-COVID

The medical management of post-COVID is a major challenge for healthcare systems worldwide and requires multidisciplinary approaches with a central involvement of primary care [15,16,17,18]. In Germany, in accordance with the recommendations in the S1 guideline and the report of the Federal Joint Committee (G-BA), published on 21 December 2023, which has a binding character for the treatment of patients with statutory insurance, the GP plays a central role in coordinating care, acting as the main point of contact for patients throughout the care process [8,19]. In addition to the G-BA [8] recommendations, studies from the patient perspective also demonstrate that GPs play a key role in post-COVID care [20,21].

Current care presents a number of challenges for GPs. These include the diversity of symptoms, the difficulty in differentiating the condition from other diseases, and the lack of knowledge about the condition [22,23,24,25]. Further difficulties are encountered in the recording of symptoms, the lack of comprehensive guidelines and treatment plans, and the time-consuming nature of diagnosis [23]. In addition, long waiting times to consult specialists and limited health insurance budgets for diagnostic tests and therapies have constrained the care of patients after a COVID infection [23,26].

Some post-COVID patients express concern that their symptoms, which restrict them in their daily lives, will not improve [20,21,23,27]. Therapies that patients feel are necessary are often not prescribed because the burden of disease is deemed to be too low [20,21]. They also expressed feelings of helplessness, insecurity, being left alone, and not being taken seriously by doctors and in their work and home environments [18,20,21,22,23,27,28]. Additionally, patients were also concerned about their own ability to work [21,23,27].

Patients want a structured approach and coordination of their medical care and specific services to improve their health [22]. It is of significant importance to patients that all symptoms are assessed through a multidisciplinary approach, with the provision of listening and psychological support [21,29]. Furthermore, it is crucial that healthcare providers demonstrate empathy and that contact persons with the necessary competencies are made available to those affected [20,22,29]. 

### 1.2. State of Research and Objective

As indicated by experts, structured care for post-COVID patients is of significant importance [19,20]. In Germany, in accordance with the G-BA guideline, this includes drawing up a treatment plan based on the results of a basic assessment, considering suggestions for adapting and updating the plan, and the furnishing of information to patients [8]. In addition to the G-BA [8], the National Institute for Health and Care Excellence in the United Kingdom (NICE) [15] also recommends, among other things, the promotion of self-management in the healthcare of people with post-COVID. 

In previous studies, case management has been used as a means of providing structured care for patients with chronic conditions. Patients reported positive experiences, including regular contact [30], continuity of care [30], improved access to specialists [31], and enhanced communication with healthcare providers [30,32,33]. Furthermore, patients reported increased awareness of their illness [32], a sense of being taken seriously [32], a trusting relationship with their carers [30], and involvement in decision-making [31]. However, patients identified a lack of understanding of case management [31] and the sometimes mechanical processes as weaknesses [32].

Despite the existence of several national and international guidelines for post-COVID care, it remains unclear how these recommendations can be implemented in specific healthcare settings. Furthermore, it is unclear whether introducing case management in the context of post-COVID would improve care. To address these challenges and to strengthen the role of GPs, in the PostCovidCare pilot study (PCC), a case management application with a browser-based symptom diary was developed based on existing software in 2021–2022. This was piloted in GP practices (September 2022 – March 2023) in Baden-Wuerttemberg, a federal state in the south of Germany [34,35].

The aim of this study was to explore patients’ experiences with the PCC intervention. A qualitative design was chosen to capture the complexity and diversity of individual experiences and to generate insights that can inform future interventions in person-centered post-COVID care.

## 2. Materials and Methods

In order to address the research objective above, qualitative semi-structured interviews with patients were conducted as part of the process evaluation of the PCC pilot study. This study was approved by the Ethics Committee of the Medical Faculty of the University of Heidelberg (S-165/2022). The COREQ (Consolidated criteria for Reporting Qualitative research) checklist was applied in the preparation of this paper (Appendix A: COREQ-List2) [36].

### 2.1. Implementation of the Intervention as Part of the PCC Pilot Study

The primary objective of the PCC module, developed as part of a feasibility study within the CareCockpit software, version 3.7 and 3.8 [35], is to enhance the role of GPs in the care of individuals with post-COVID through the utilization of software-assisted case management with a browser-based symptom diary. The content encompasses the provision of a framework for the medical care process, the dissemination of information, the integration of a medical consultation, and the creation and regular review of an action plan. Furthermore, the documentation of symptoms should empower patients and involve them in the management of their health condition [34].

In terms of health care providers, GPs and their specialized medical assistants from GP practices participating in the GP-centered care program in Baden-Württemberg were included. Patients were included by their GP. In terms of the patient population, this study included consenting adult patients who had been diagnosed with post-COVID, as defined by the assignment of ICD U09.9, and who exhibited at least one associated coded symptom and a positive SARS-CoV-2 detection. Additionally, participants were required to meet technical specifications and demonstrate willingness to utilize a browser-based symptom diary [34].

The use of the PCC module included enrollment, access to the browser-based symptom diary, an appointment for assessment, and at least one follow-up between assessment and the final follow-up assessment at three months [34].

### 2.2. Study Design

Qualitative, guided individual interviews were seen as an appropriate methodology for addressing this research question, as they permit the collection of experiences and perspectives from the patients [37]. This approach can provide exploratory insights into the respondents’ lived experiences [37]. The qualitative approach ensures an openness to detailed insights [37]. Furthermore, as part of the process evaluation, interviews are an appropriate method for determining the experiences of participants with the intervention, with the objective of deriving insights for the further development of the intervention [38].

### 2.3. Recruitment of the Interview Participants

The 33 patients enrolled from the 10 GP practices in urban and rural areas in Baden-Wuerttemberg who participated in the PCC pilot study were invited once to participate in an interview as part of the process evaluation at the end of the intervention. Letters were sent by post to participating practices in May 2023, asking them to send the interview study information to patients included in the PCC pilot study. If the study participants indicated their willingness to participate in this study, they were contacted by telephone to discuss data protection and the aims of the interview. Participants received €50 for their participation in the interview.

In order to participate in the qualitative interviews, it was necessary for the subjects to have provided written consent to be recorded. Participants were informed in advance about the voluntary nature of participation and their right to withdraw at any time without consequence. They were also explicitly informed that they could pause or terminate the interview at any point without providing a reason.

### 2.4. Data Collection

The interview guide (Appendix A: Interview Guide) was developed by the study team based on the RE-AIM framework and subsequently revised following discussion and feedback at an interprofessional research workshop. Following the initial collection of demographic data, the topics covered in the guide included the following: (1) the extent to which the target group has been reached (reach), (2) the manner in which the intervention is delivered (implementation), (3) expectations, (4) barriers and disappointments (adoption), (5) the negative and positive effects of the intervention (efficacy), and (6) the long-term use of the tool (maintenance) [39].

As a result of the specific questions relating to the intervention, the first interview with a participant in the pilot study was used to pilot the interview guide. Subsequently, minor adjustments were made to enhance the clarity of the interview questions. The pilot interview was included in the data analysis due to the hesitant feedback received when recruiting patients for the data analysis via the participating GP practices.

All interviews were conducted via telephone in German by the first author and recorded on a digital audio device with the written consent of the patients. Subsequently, the interviews were transcribed using the transcription software noScribe (Version 0.3) [40] and then re-checked. All information that could potentially identify the individual or the practice was anonymized. 

The interview quotations and the guidelines were translated by the first author into English for the purpose of this publication.

### 2.5. Data Analysis

Following the initiation of the textual analysis and the creation of case summaries for each interview, the data analysis was conducted using a qualitative content analysis approach, as outlined by Kuckartz [37,41,42].

Initially, four interviews with a maximum of contrasts were coded openly (inductively) [41]. Following the coding of the interviews, the codes were summarized and grouped together, and main categories were formed [41]. These were subsequently modified in light of the findings from the subsequent interviews, which were also subjected to coding [41]. Thematic saturation was considered reached when the category system proved stable across several consecutive interviews and no further new codes were needed to capture relevant aspects from the data. This process was repeated until the point of data saturation was reached. Once the coding process was complete, a codebook was created [41] (Appendix A: Code Book). In parallel, summaries of the coded text passages pertaining to a given code were produced in parallel [42,43]. The MAXQDA software [43] (version 2018) was employed to analyze the data.

Due to the identified thematic saturation and the consistency of the number of conducted interviews with findings from qualitative research, no reminder was sent, and the analysis was terminated [44].

All interviews and analysis were conducted by the first author. While single-person coding can introduce bias, efforts were made to mitigate this by regularly discussing the coding and interpretation with members of the research team (C.U. and S.S.) and presenting the evolving analysis in research workshops for critical reflection. The analysis was influenced by the RE-AIM framework, but the final categorization system was not primarily based on the framework. Neither the transcripts nor the initial findings were shared with respondents for feedback. 

## 3. Results

Of the 33 patients who participated in the pilot study, *n* = 10 were willing to be interviewed. For data protection reasons, the project team does not know at which practice the patients were treated. An interview was conducted with all those who were interested. The interviews took place between July 2023 and – September 2023 and had an average duration of 30:28 min. One interview was continued on the same day with a short interruption due to an overload situation of the interviewee. The author conducted the interviews in her home; no other person was present during the actual interview except the interviewer and the interviewees. The interviewer had no other relationship with the interviewees. All interviews conducted with the patients were used for the analysis. Table 1 shows the demographic data of the respondents. 

Interviewees’ accounts of their experience of the intervention and their evaluation of it varied widely across interviews. From the patients’ perspective, it was not always possible to identify whether a task was carried out by a specialized or a regular medical assistant. Therefore, the umbrella term medical assistant is used in the results report. Two patients reported that they had not been seen by their usual GP but had switched either within a practice with two or more GPs or to another GP practice participating in the pilot study. As the other patients did not mention this, it was assumed that they were being cared for by their regular GP. 

The structure of this chapter is presented by Table 2:

### 3.1. Conducting Questionnaires and Tests

This encompasses the patients’ experiences of completing questionnaires and carrying out physical and cognitive tests. Additionally, it includes the performance of physical and cognitive tests.

A large proportion of respondents described that they were given some documents to complete before the first doctor’s appointment planned as part of the pilot study and were told to take this home with them. Most of the respondents were asked to complete questionnaires as "homework" (PAT_4). As the patients reported, the content was about their current state of health. The questionnaires were usually collected from the practice, although one patient reported having received them by post. In addition, patients described their experiences of completing questionnaires at the medical practice. About half of the participating patients sat in the waiting room of the GP practice or were provided with a separate room for this purpose. 


*Yes, they gave me these, how do you say, documents, they gave them to me and then they said, yes, now I can sit outside and fill them out in my own pace for as long as I need […].*

*(PAT_7)*


The patients reported that this task was mostly carried out by medical assistants. When there were questions, respondents mentioned that they had received support from the medical assistants. Individual patients also reported that they had completed questions on the computer together with the medical assistant.

The extent to which the symptom diary or other questionnaires were meant here remains unclear, but the text passages were nevertheless assigned to this code, as the topic was the completion of questionnaires. 

Two patients found answering the questionnaires exhausting and demanding at times. 


*In the beginning, that was very exhausting. Answering all the questions and everything and, yes, then sitting outside and filling out the questionnaire, then going back inside, then back at the computer again, answering questions and so on. It was exhausting, I have to be honest. Yes, because my perseverance is still suffering a bit at the moment.*

*(Interview PAT_7)*


A minority of those affected gained insights and made positive associations with the help of the questionnaires; other patients found it difficult to evaluate. 

In addition to the questionnaires, the patients reported that a physical examination had taken place. These included vital signs such as blood pressure, blood sugar, and oxygen saturation, as well as imaging procedures such as abdominal ultrasound and tests of physical resilience. In some cases, cognitive tests were also reported.

The interviewees rated the physical examination as positive and informative. 


*[…] but as I said, the first one was okay with the examination, really a thorough physical examination, so I’ve never been so thoroughly examined by a doctor, I have to say quite clearly, I thought it was nice, […].*

*(Interview PAT_1)*


Moreover, another patient felt that the examinations were nothing special. From her point of view, they were rather superficial and paid less attention to individual symptoms and needs. 

The patients reported that the GPs carried out the physical examinations. According to the patients, the activities of medical assistants in the GP practice included carrying out questionnaires and taking vital signs.

### 3.2. Doctor–Patient Consultation with the GP

This section reports on patients’ experiences with content and communication during doctor–patient consultations, especially regarding health status, symptom diaries, action planning, and the provision of information materials. In the following section, the findings are organized by patient engagement, emotional response, and perceived coordination of care.

Respondents reported being asked about their state of health during the consultation, symptoms, complaints, and general condition dealing with the illness and the development of their health: 


*[…] then he just wanted to know how I was doing, with COVID, what my current complaints are, […].*

*(Interview PAT_1)*


Half of the interviewees indicated that their questionnaires were discussed again with their doctor after they had completed them. 


*He went through what I had prepared, i.e., the forms that I had filled in, and we discussed them again together […].*

*(Interview PAT_2)*


Some patients felt insufficiently engaged. For example, a few interviewees said that their data were not discussed or that they were unsure about the intended use of the forms. One patient mentioned receiving no follow-up consultation despite her GP promising to get back to her with questions.

In addition, experiences with symptom diaries varied. The majority of the patients brought it to the consultation and discussed it with their GP. 


*And then I always brought it to him. He looked through it and then we talked about it again briefly.*

*(Interview PAT_10)*


Moreover, other interviewees reported that the symptom diary was not discussed with the GP. Individual patients were unsure if the diary was even intended to be handed in. Patients also made different accounts of whether they had to hand in the symptom diary or not. One patient expressed a lack of understanding that she was not supposed to hand in the completed forms at the GP practice. 

Many of the interviewed patients also reported the creation of an action plan for further care of their post-COVID illness as well as referrals to specialists and prescriptions for therapies (e.g., respiratory, physiotherapy, and occupational therapy) or diagnostic procedures, including magnetic resonance imaging (MRI), computed tomography (CT), and measurement of brain waves. Patients in the pilot study also reported receiving recommendations for medications, physical exercises, audiobooks, and apps, and tips on how to behave in difficult situations. 


*And the GP gave me the suggestions that this study made for my symptoms, especially at the first appointment.*

*(Interview PAT_10)*


The action plan and the usefulness of the prescribed therapies were evaluated at the follow-up assessment appointments. This was used to determine whether, for example, a further prescription would be useful. According to the patients, the results of the tests with medical specialists were discussed during consultations with the GP that took place as part of this study. 


*Yes, exactly how useful the [therapies] were and whether it made sense and whether I would like to do it again, or exactly what had improved.*

*(Interview PAT_2)*


Nearly half of the interviewees reported receiving information material at their GP practice. For instance, a list of therapists, information on symptoms, audiobooks, breathing exercises, and the "*post-COVID flyer*" (Interview PAT_6). 


*I didn’t receive much in the way of information material beyond the treatment and discussions we had.*

*(Interview PAT_10)*


The reactions towards the information material ranged from filing them in folders to the fact that the interviewees found the information they received to be not adequate for their situation and did not use it. 

The patients reported that the consultation was conducted primarily by the GP, with medical assistants in the GP practice occasionally participating or assuming responsibility for specific elements of the consultation. 

Overall, nearly half of the interviewees rated the doctor–patient consultation as useful and the information obtained as helpful. In particular, the receipt of referrals to specialists and therapeutic measures. The participants valued being taken seriously and the GP’s time and interest in improvement in the patient’s situation. 


*It was good in itself, because it helped me a lot, especially with the pulmonologist […]*

*(Interview PAT_4)*


However, negative aspects were also mentioned, including the design of the PCC module and the associated high time investment for GPs. Furthermore, the content of the consultation was perceived as duplicating that of the medical assistants. Three patients perceived no added value of the doctor–patient consultation compared to a “normal” consultation. 

A minority of the interviewees indicated insufficient involvement and a lack of explanations regarding the examinations conducted, therapeutic concepts, the results of completed questionnaires, or the subsequent steps. One patient perceived this as having consequences for her application for social benefits:


*And if I don’t get any diagnoses[…] I simply don’t have [them] if it’s not taken into account.*

*(Interview PAT_8)*


Therapeutic measures were seen as beneficial and hopeful; the impact of the program on their own health was seen as divergent. Some patients reported significant effects, while others were unable to notice any change in their condition. 

Two patients classified recommendations for action regarding the personal environment as easy to realize. Furthermore, the interviewees encountered limitations due to personal interests and structural restrictions. 

### 3.3. Symptom Diary

This section presents patients’ statements on the use and receipt of information on the symptom diary and their assessments of it. 

The majority of patients utilized the paper-based form of the symptom diary (Table 1). According to the interviewees, one reason for the increased use of the paper-based symptom diary was the delayed availability of the online diary. Even after it was made available, patients mentioned technical problems that led to the offline version being used. One patient expressed her lack of understanding for this. Nevertheless, some interviewees had the opinion that both options were justified and made sense depending on the patient’s needs and technical skills. According to the interviewees, the online version offered the advantage of being able to complete the form from any location, avoid losing documents, and transfer the data to the GP without any problems.


*Yes, well, it didn’t work at first and then I tried it again in between, but it didn’t really want to either. And then, then I stuck with the paper and then, because in the end it’s not really that much different.*

*(Interview PAT_10)*


The participants reported that the frequency with which they completed the diary ranged from twice a day to twice during the entire intervention period. In some cases, fixed times of day were set for this, or a time shortly before the next appointment at the GP practice was used to complete the symptom diary. One patient reported that she no longer filled in the symptom diary after a certain point, as it was not collected by the practice staff at the GP practice. 


*So I did it every three days at most, every three to four days, if there was a day when I didn’t do so much, then I didn’t have to enter anything or if there was something going on, then I realized, so I always did something every three days at most.*

*(Interview PAT_5)*


In addition, respondents indicated that they were uncertain about the frequency with which they should complete the symptom diary and the subsequent steps that are taken following the submission. This lack of clarity led to confusion among a majority of the patients. 

Two interviewees found the symptom diary to be beneficial. One patient described the symptom diary as ineffective because she could not gain any insights from it without consulting her GP. Furthermore, two patients perceived the symptom diary to be beneficial only if it was completed and made available to the GP. In particular, the awareness of their own illness was perceived as a positive effect by nearly half of the patients who completed the symptom diary. However, for one patient, negative associations were more prevalent. Some interviewees described the scope and comprehensibility of the questions within the symptom diary as a hindrance, whereas one patient commented positively on this aspect. 


*So I have now found the symptom diary good for me personally for reassurance. […] That was really instructive for me (3).*

*(Interview PAT_9)*


### 3.4. Patients’ Overall Experience of the Intervention

During the interview, the interviewees mentioned aspects that could not be assigned to the individual intervention components but rather related to the external circumstances affecting the intervention as a whole. These were identified as relevant to answering the research question and are therefore presented below. 

The respondents described making appointments as uncomplicated. In particular, patients with a short distance to the GP practice mentioned the advantage of being able to drop by spontaneously, for example, to take vital signs. The majority of interviewees mentioned the short waiting times and a quick and structured process. Others described sometimes long waiting times due to technical problems or a perceived lack of organization at the GP practice. One patient reported separate appointments with the medical assistant in the GP practice and the GP. According to the interviewees, the appointments within the intervention period differed mainly in terms of length. The majority of interviewees felt that the first appointment was the longest. The reasons given for this were the limited experience with the process and technical problems. The interviewees identified no differences at all or only some differences in terms of the content of the appointments. 

Overall, the appointments were perceived as time-consuming. In addition to the personal effort required of the interviewees, the GP’s effort was also emphasized. On the one hand, the appointments were perceived as too long or even stressful; on the other hand, the resulting closer relationship with the GP was emphasized by one respondent. 


*But it’s somehow become a bit closer. Simply because of the study. Because we kept talking about it.*

*(Interview PAT_6)*


Overall, the participants mostly rated the intervention positively. For example, the participants highlighted the support, the regular appointments, the prescription of therapies, and also the insights that the participating GPs were able to gain from the intervention. However, two patients reported negative views. One of the two patients was only able to mention the prescription of therapies as a positive effect. Two patients rated the intervention as neutral.


*No, neither one nor the other, it just went along passively, I have taken off for the three appointments and that was okay.*

*(Interview PAT_4)*


Finally, the patients mentioned the receipt of therapies, referrals to specialists, and the prescription of medication. However, they did not categorize this as a change in their healthcare. Two patients, however, emphasized the receipt of therapies as a change in their care.

### 3.5. Future of the Intervention

In order to be able to derive implications later, the patients were asked what they would like to see in the future of the intervention and what would need to be adapted.

Overall, the patients required more support for the care of their post-COVID disease as part of the intervention. However, what exactly was meant by this was not specified further. For the future, patients wished for more information in the form of physical information materials about their post-COVID disease. It is important to note that the patients expressed a preference for information from reliable sources. They also indicated a desire for explanations of therapeutic concepts and a website with information. In addition to the desire for information, the interviewees expressed a preference for relaxation and mindfulness exercises that they could perform at home and support in finding therapy places as an improvement for the future.


*Yes, that you might also get other support somehow […].*

*(Interview PAT_3)*


With regard to the continuation of the intervention, the patients wanted documentation of the symptoms and experiences with therapies by and for post-COVID sufferers. It is therefore recommended that the focus should be placed more strongly on the individual symptoms of patients within the care program. In addition, the testing of physical stress and the expansion of appointments were suggested as improvements. 


*I don’t know if you could divide the study into different parts […] in order to go into the individual cases more specifically […].*

*(Interview PAT_4)*


When asked whether patients would recommend participation to others, the majority of interviewees responded positively. The interviewees cited the benefits for their own healthcare, such as receiving treatments and examinations, as reasons for participating. Moreover, this could facilitate the development of novel treatments, enhance management strategies, and increase societal acceptance of the post-COVID syndrome.


*Because I think that’s the only way to get more information and the only way to find the perfect therapies or treatments. So, doing nothing simply doesn’t help anyone.*

*(Interview PAT_2)*


In this context, the time required was sometimes described as minimal. Conversely, other patients emphasized that the substantial time investment would be a reason not to recommend participation in this study. 

## 4. Discussion

The experiences made by the participants with the study intervention implemented in 2022–2023 to improve the care of patients with post-COVID were heterogeneous due to different implementation and different evaluations of the intervention. The process was described as predominantly structured. It involved a lengthy assessment appointment, follow-up appointments, questionnaires, support from medical assistants, physical examinations, diagnostic tests, and consultations between doctors and patients. In contrast to the implementation of tests and questionnaires, the doctor–patient consultation played a more prominent role in the interviews. Here, participants reported that doctors actively asked about symptoms, limitations, and progression. They received an action plan with information, referrals, therapies, and behavioral recommendations. Follow-up examinations were conducted to reassess these measures. Participants found it positive to be taken seriously, to have time and interest shown, to receive support, and to feel that their relationship with their GP had become closer. Reported negative aspects included the time taken, resulting exhaustion, duplication of efforts, and insufficient involvement in the consultation process. In some cases, effects on health were described. The symptom diary was mostly used in the paper-based version due to technical issues. Its utility was generally considered to be useful if the entered data or completed forms were discussed in the doctor–patient consultation. The participating patients generally rated the intervention of the PCC pilot study positively, which was also reflected in a high recommendation rate to other patients. Patients reported changes in their healthcare, although some of these were not seen as significant changes in the eyes of the patient. However, they would like to see the range and support expanded and tailored more closely to individuals in the future. 

### 4.1. Comparison with the Existing Literature

To our knowledge, this is the first study that explores the experiences of primary care-based case management for post-COVID patients. Apps have been developed to support post-COVID patients and implementation of recommendations. However, they do not include a stepped care approach or a focus on strengthening the role of general practitioners [8,15].

The respondents in this study expressed a sense of being taken seriously, which contrasts with the negative experiences of patients in standard care [20,22]. Additionally, the extra time within the appointment reported by patients in this study can also be found in the findings of the meta-analysis by Askerud and Conder [30] on the care of patients using case management. The promotion of a closer relationship with primary care providers, as reported by the interviewees, can be supported by other studies [18,33]. 

Finally, the doctor–patient consultation and the role of GPs were also emphasized due to the receipt of referrals and prescriptions for therapies. This addresses the problem of often not receiving therapies [20]. This also aligns with the international recommendations of primary care-based post-COVID care [18,22,45,46,47]. 

In addition, patients would have liked an even more comprehensive range of services and assistance than provided in this study. Additionally, a number of patients participating in the interviews indicated that the appointments were a source of considerable stress. A review of international studies shows stress intolerance and an increase in symptoms after exertion, known as post-exertional malaise. This symptom is commonly reported in the cohort of post-COVID patients and also contributes to the diagnostic criteria of myalgic encephalomyelitis/chronic fatigue syndrome (ME/CFS) [48,49].

Previous studies have demonstrated the efficacy of case management in activating participants [33,50,51]. Additionally, patients reported being actively involved in decisions in other studies [50]. Concurrently, NICE [15] recommends shared decision-making for the planning of further care for post-COVID patients. The interviews revealed that, in some cases, the results of the questionnaire designed to inform the consultation were not discussed adequately. Some patients also felt that they had not been sufficiently involved in the consultation process. However, the reason why such experiences were only reported by some of the participants in the interviews remains unclear. This discrepancy between the intended goal of active patient involvement and the perceived situation may result from structural barriers in the software, limited time resources for GPs, competing priorities for complex diseases during the consultations, and the lack of integration of shared decision-making prompts into the software interface. Furthermore, this study was conducted in the early post-pandemic era, which presented limitations in terms of the timeframe and unfamiliarity with the clinical picture. This era enhanced the idea of rapid implementation [52]. 

From the patients’ point of view, the GPs play a central role in the intervention in the results, whereas the specialized assistant in the GP practice and their area of responsibility were seen as less relevant in the care process. In contrast, a secondary data analysis by Forstner et al. [53] indicates that the introduction of case management has an effect on the relationship between patients and specialized assistants in the GP practice and is associated with a higher intensity and an increase in trust. The qualitative study by Hoffmann et al. [54] demonstrates that specialized assistants in the GP practice play a central role in patient care. In international studies, medical assistants also play a greater role in the provision of care based on case management [30,33,50]. One possible explanation for the differing structures is that the primary objective of the PCC pilot study was to strengthen the role of GPs [34]. The specialized assistant in the GP practice was not explicitly mentioned in this context and was intended to support the GP in the implementation. It was therefore to be expected that the focus would be on GPs. Another reason could be the difficulty of delegation due to a low level of evidence and the dynamic change in knowledge regarding post-COVID disease [29]. In the context of the impending shortage of doctors in Germany, however, the question remains open as to whether this approach is indeed the most effective or whether medical assistants can and should be more involved in the provision of care [55,56].

The use of the online symptom diary was not possible for the majority of respondents due to technical problems. In some cases, the diary was also perceived as irrelevant or was not discussed at all by the GP, leading to a sense of futility among patients. Nevertheless, previous studies posit that digital symptom recording is less prone to error than a paper-based version [57,58]. The symptom diary in general—regardless of the format—was found to be a useful tool in the present study, provided that the results were made available to GPs and discussed. However, this was the case for only a small number of patients. This highlights a critical implementation gap and points to broader challenges in digital health adoption in primary care settings. In particular, it raises questions about how digital tools are introduced, integrated into routines, and perceived by both patients and providers. Furthermore, there is a convergence of findings between the present study and previous research on the enhancement of patients’ awareness of their own health status [59]. The symptom diary could therefore confer benefits for patients and carers if implemented in an appropriate manner.

The findings resonate with broader debates on digitalization in primary healthcare, particularly the need for usable, meaningful tools that support—not burden—patients and providers [60]. The results underscore that patient empowerment through digital tools requires not only access but also contextual integration, feedback loops, and adequate training [61,62].

### 4.2. Strengths and Limitations

A major strength of this study is that it represents an early evaluation of a rapidly implemented pilot intervention designed to improve the care of post-COVID patients. A representative sample was successfully recruited that is comparable to those of other studies in terms of gender and age [21,28].

The results cannot be generalized due to (1) selection bias, as it can be assumed that the participating GP practices already have an interest in the care of patients with post-COVID; (2) the sample is limited to patients who were able to visit the GP practice on-site, thus excluding housebound patients; and (3) the results presented here are limited to the qualitatively collected perspective of post-COVID patients, without considering the perspective of practice staff or quantitative data. 

The intervention was implemented in a real-world setting, with some unavoidable challenges due to the heterogeneity of GP practices [63].

The technical issues associated with the online symptom diary limit a comprehensive evaluation of this aspect through patients’ experiences. 

The interval of a minimum of four to a maximum of nine months between the end of the intervention and the interview may contribute to recall bias, particularly in patients experiencing cognitive impairments as part of their post-COVID condition [7]. But it also allows the integration of a long-term perspective. Potential social desirability bias was countered by using a person who was independent of the intervention.

In terms of methodological limitations, the qualitative interviews and the inductive coding of the material were both conducted exclusively by a single researcher. Therefore, it is not possible to fulfill the criteria of intra- or intercoder agreement [42]. In order to minimize the risk of bias and to maximize the reliability of the data analysis, a continuous exchange and reflection was held with the internal project coordination team, and discussions in the research workshop took place. Moreover, respondent validation (member checking) was not conducted, which may limit the interpretative validity of the findings.

### 4.3. Implications for Research and Practice

The results of this pilot study provide initial insights into understanding of how the piloted intervention works from the patient’s perspective [64]. The findings can be used in the development cycle of complex interventions by integrating them into further development. Further studies should complement the reported patient’s perspective by integrating the perspective of the practice staff of the participating practices, as well as quantitative data. 

The results of this study already provide some suggestions for improving the project, such as using telemedicine options for consultations in appropriate cases to reduce stress and include homebound patients. Based on the results, there should be a greater focus on implementing shared decision-making and individualized care. It would be beneficial to involve patients in the planning and development of future developments and to strengthen the involvement of specialized medical assistant staff in the GP practice in order to relieve the burden on GPs. To make profitable use of the symptom diary, future applications should simplify it to improve user-friendliness and patient compliance and ensure that discussing the entries is an integral part of the treatment process.

On a systemic level, this study highlights the need for enhancing competencies for GPs. In addition to further developing post-COVID case management, this could also include training programs in this theme as well as about digitization as part of continuing medical education [65]. Additionally, improvements in IT infrastructure within GP practices are crucial to support the seamless use of digital interventions such as software-based case management, ensuring technical reliability and accessibility for both patients and providers [66].

## 5. Conclusions

The PostCovidCare pilot study conducted at an early stage of the post-COVID era demonstrated the basic feasibility of case management in a primary care setting from the patients’ perspective. The patients’ experiences in this study indicate that relevant patient needs explored in previous studies—such as the doctor–patient consultation, the time received, and the therapies received—could be positively influenced by the intervention implemented in the pilot study. Furthermore, additional aspects were identified that require consideration when developing the intervention further. For future projects, it is important to integrate patients into the design from the outset in order to ensure more individualized and patient-oriented care and continue to adapt the software to users’ needs. In conclusion, the pilot study provided valuable insights into how future interventions in person-centered post-COVID care, including case management, could be further developed within post-COVID care structures. However, a final evaluation of the pilot intervention cannot be made without considering the perspectives of GPs and medical assistants in GP practices as well as triangulation with quantitative data. 

## Figures and Tables

**Table 1 healthcare-13-01701-t001:** Demographic data of the patients participating in the telephone interviews (*n* = 10).

Patients (*n* = 10)		
Gender	Male	3 (30%)
	Female	7 (70%)
Age	<40 years	3 (30%)
	>40 years	7 (70%)
Place of residence	City/near a city	2 (20%)
	Rural	8 (80%)
Symptom diary ^1^	online	2 (20%)
	Paper form	8 (80%)
Ability to work	Yes	6 (60%)
	No	3 (30%)
	unclear	1 (10%)

^1^ At the beginning of the PCC pilot study, a paper-based symptom diary was integrated into the intervention due to technical difficulties with the online symptom diary.

**Table 2 healthcare-13-01701-t002:** Themes and subthemes.

Theme	Subtheme
Conducting questionnaires and tests	Patients’ experiences of completing questionnaires and tests Emotional Impact and Perceived Value
Doctor–patient consultation with the GP	Discussion of Health Status Discussion of Symptom Diary Action plan Information Materials Emotional Impact and Perceived Value Perceived Impact of the Program
Symptom diary	Patients’ experiences of completing questionnaires and tests Emotional Impact and Perceived Value
Patients overall experience of the intervention	Organizational aspects Emotional Impact and Perceived Value
Future of the intervention	Recommendations for the future Recommendation to other patients

## Data Availability

The datasets generated and/or analyzed during the current study are not publicly available because of assured data protection regulations. The raw data supporting the conclusions of this article will be made available by the corresponding author upon request.

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
