# Peer review of "Patient Experience from a Pilot Study Implementing Software-Based Post-COVID Case Management in GP Practices—A Qualitative Process Evaluation"

_healthcare, 2025, doi:10.3390/healthcare13141701_

Round 1
Reviewer 1 Report
Comments and Suggestions for Authors
General Overview
This article presents a qualitative evaluation of a software-assisted case management intervention for post-COVID patients in general practices in Germany.
While the study addresses a highly relevant topic and applies appropriate qualitative methods, there are several areas that require refinement in terms of methodology, interpretation, clarity, and critical depth.
Section-by-Section Review
- Title and Abstract
Strengths:
- The title accurately reflects the study focus and design.
- The abstract captures the key themes of the findings.
Weaknesses:
- Abstract contains grammatical issues and vague phrasing.
- The conclusion of the abstract is weakly formulated.
Suggestions:
- Improve clarity and grammar in the abstract (as detailed earlier).
- Strengthen the final sentence to highlight the study’s contribution and implications more powerfully.
- Introduction
Strengths:
- Provides comprehensive background on post-COVID syndrome and the challenges faced by GPs.
- Cites relevant WHO and NICE definitions and guidelines.
- Justifies the need for structured, GP-centered interventions.
Weaknesses:
- Slightly too long; could be more concise.
- Repeats some information unnecessarily (e.g., prevalence and symptom variety).
- Weakly transitions from literature to the specific intervention studied.
Suggestions:
- Shorten redundant sections about symptom prevalence.
- Clearly delineate what is known and unknown in the field.
- End with a sharper articulation of the study aim and rationale for using a qualitative design.
- Materials and Methods
Strengths:
- The RE-AIM framework and COREQ checklist are cited, which lends structure and transparency.
- Interview guide development and data collection are clearly described.
- Inclusion criteria are specific.
Weaknesses:
- The role of the first author as both interviewer and coder is potentially problematic for bias.
- Only 10 out of 33 patients participated, raising concerns about representativeness.
- No triangulation, intercoder reliability, or respondent validation. How do you maintain the quality of the analysis?
Suggestions:
- Acknowledge and discuss potential biases due to single-person coding and interviewing more explicitly in this section.
- Mention how thematic saturation was assessed (i.e., why 10 interviews were deemed sufficient).
- Explain how patient confidentiality and emotional well-being were protected during interviews on a potentially distressing topic.
- Results
Strengths:
- The results are rich and categorized thematically with illustrative quotes.
- Several important themes are explored: the GP consultation, symptom diary, stress of appointments, lack of involvement, and therapy outcomes.
- The patient voice is prominent, which is appropriate for qualitative research.
Weaknesses:
- Overlap between categories (e.g., “consultation” vs. “action plan” vs. “appointment experiences”) reduces clarity.
- Quotes are sometimes long and not well-integrated into analytic narrative.
- Descriptive rather than interpretive: little analysis of why experiences differed or what structural factors contributed.
Suggestions:
- Merge overlapping categories and present themes in a more cohesive structure (e.g., group by patient engagement, emotional impact, care coordination).
- Synthesize patterns more clearly: What types of patients tended to feel more positive? Were there differences by gender, diary type, or GP continuity?
- Add more interpretive commentary to move beyond surface-level description.
- Discussion
Strengths:
- Connects findings with relevant literature, particularly in relation to patient experience, case management, and GP roles.
- Appropriately acknowledges that the intervention improved some aspects (e.g., patient-GP communication).
Weaknesses:
- Tends to restate results rather than critically analyze them.
- The discussion of medical assistants' limited perceived role is superficial, despite its importance.
- The lack of patient co-design is noted, but not explored in depth.
Suggestions:
- Provide a deeper critical reflection on why shared decision-making was lacking, despite being a program goal.
- Discuss whether the GPs’ high workload, structural constraints, or the design of the software may have limited engagement.
- Address the finding that symptom diaries were often unused or ignored—what are the implications for digital health adoption?
- Link more strongly to broader debates about digitalization, patient empowerment, and primary care transformation.
- Strengths and Limitations
Strengths:
- Transparent about sample size, lack of triangulation, and real-world challenges.
- Notes the strength of early evaluation and representative demographics.
Weaknesses:
- Some limitations are underdeveloped (e.g., no discussion of potential interviewer bias).
- Selection bias is mentioned, but not connected to implications for interpreting findings.
Suggestions:
- Expand on methodological limitations: single coder/interviewer, absence of respondent checking, recall bias especially in cognitively impaired post-COVID patients.
- Clearly state that findings are not generalizable.
- Implications and Conclusion
Strengths:
- Notes that future development should include patient involvement and more tailored care.
- Recommends exploring telehealth and delegating more to non-physician staff.
Weaknesses:
- Somewhat generic and unambitious recommendations.
- No real call to action for policy or practice audiences.
Suggestions:
- Offer specific design recommendations for future interventions (e.g., simplify symptom diary; ensure regular GP-patient discussion of entries).
- Advocate for systemic changes, such as training for GPs, or IT infrastructure improvements.
- Strengthen conclusion by summarizing how this study fills a research gap and what future research should address.
Visuals and Supplementary Materials
Strengths:
- Use of supplementary materials (interview guide, COREQ checklist, codebook) supports transparency.
Weaknesses:
- In the main text, there are referencing errors (e.g., “错误!未找到引用源。”), which must be corrected.
- Table 1 is helpful but incomplete; lacks relevant clinical info (e.g., symptom duration/severity).
Suggestions:
- Add a brief summary of interview themes in a visual (table or figure).
- Fix reference errors and improve formatting consistency.
Overall Recommendation
Recommendation: Major Revisions Required
This study is a valuable contribution to the growing literature on post-COVID care and digital case management in primary care. However, it currently lacks the analytical depth and methodological rigor expected in qualitative health research. Substantial revisions—especially in thematic analysis, discussion, and conclusion—are needed to improve its impact and clarity.
Author Response
Dear reviewer,
Thank you very much for your comments. We have thoroughly revised the manuscript and uploaded a detailed response in the PDF.
Best regards,
The authors

Reviewer 2 Report
Comments and Suggestions for Authors
Please, see the attached document.

Author Response

(The authors gave the same response as above.)

Reviewer 3 Report
Comments and Suggestions for Authors
This article presents a qualitative evaluation of patients' experiences with a post-COVID case management intervention implemented in primary care settings in Germany. The intervention comprised structured consultations with GPs, support from medical assistants, and the use of a symptom diary in either digital or paper-based format. Semi-structured telephone interviews were conducted with ten participants from the pilot study. Findings revealed heterogeneous experiences: patients appreciated the time dedicated by their GP, active listening, and access to therapies, but criticised the lack of involvement in decision-making and technical challenges associated with the digital diary.
Overall, the article demonstrates a clear commitment to adhering to methodological best practices in qualitative research. The explicit reference to the use of the COREQ checklist is, from the outset, a positive indicator. The methodology is described systematically, the data collection process is transparent, and the results are presented with rigour, including illustrative quotations from participants to support the thematic analysis.
In the paragraph beginning with “In previous studies case management has been used as a means of providing structured care for patients with chronic conditions...”, multiple specific claims are presented, such as improvements in continuity, access to care, communication, patient empowerment, and decision-making. However, all these statements are collectively supported by a grouped citation ([31–34]) placed at the end of the paragraph. To enhance transparency and allow readers to verify the source of each specific finding, it is recommended to attribute individual references to the corresponding claims. This would improve the traceability of evidence and strengthen the scientific rigour of the literature review, ensuring that each cited study is accurately linked to the findings it supports.
While the Methods section indicates that the intervention was implemented in 10 general practices with 33 patients enrolled, it is not explicitly stated how the final sample of 10 interviewed patients was distributed across these practices. Moreover, the article does not provide details about the characteristics of the participating practices, such as geographic location (urban vs rural) or size. Given that implementation contexts can significantly influence both the delivery of complex interventions and patients’ experiences, it would be valuable to clarify: Whether the 10 interviewed patients were drawn from all participating practices or from a subset only; The diversity of the general practices involved in the pilot study. Including this information would enhance the transferability of findings and provide greater contextual depth to the qualitative analysis.
The manuscript contains two instances of unresolved reference errors displayed as "Error! Reference source not found."
It is recommended to review the original document and correct or replace the missing cross-references to ensure clarity and completeness.
Author Response

(The authors gave the same response as above.)

Round 2
Reviewer 1 Report
Comments and Suggestions for Authors
I appreciate the opportunity to review the article. I believe it is of sufficient quality to be published. Congratulations on the excellent revision work.
Author Response
Dear reviewer
Thank you very much for your valuable feedback.
Best regards
The authors
